# Multidisciplinary and Transdisciplinary Collaboration in Nature-Based Design of Sustainable Architecture and Urbanism

**Anosh Nadeem Butt** * and **Branka Dimitrijević**

Department of Architecture, University of Strathclyde, Glasgow G1 1XQ, UK
* Correspondence: anosh.butt.2018@uni.strath.ac.uk; Tel.: +44-7404-720-816

**Abstract:** Multidisciplinary, interdisciplinary, and transdisciplinary collaboration (TDC) continue to address complex societal problems such as sustainable development, global environmental change, and public health challenges. Nature-based design (NBD) methods including biomimicry, biomorphism, biophilia, bio-utilization and zoomorphism are essential for the design of the sustainable built environment (SBE). Currently, there is no transdisciplinary collaboration framework (TCF) to support the NBD of the SBE. The first step to fill this gap is through systematically exploring the applications of multidisciplinary research (MDR) in building design and by conducting a case study on the challenges to the MDR in the application of NBD methods for the SBE in the Faculty of Engineering and the Faculty of Science at the University of Strathclyde, Glasgow, UK. The systematic literature review and the survey results of academics on MDR collaboration showed a lack of transdisciplinary research (TDR) due to limited communication between disciplines. The research findings showed a lack of communication between academia and the Architecture, Engineering, and Construction (AEC) industry to advance NBD innovations for the SBE. The findings indicated that a TCF for research on NBD is needed to support knowledge exchange within academia and with industry for reducing the negative impacts of the building industry. Findings from the current research and future research will be used to develop and test a general TCF and then to develop a TCF for the NBD of the SBE aligned with the RIBA Plan of Work.

**Keywords:** multidisciplinary collaboration; transdisciplinary collaboration; nature-based design; sustainable built environment

## 1. Introduction

On 1 January 2016, the United Nations announced sustainable development goals (SDGs) as a blueprint for achieving a sustainable future in accordance with the 2030 Agenda for sustainable development. The 17 SDGs [1] address global challenges, including those relating to architecture and urbanism and the impacts of the built environment—sustainable cities and communities (SDG-11), responsible consumption and production (SDG-12), and climate action (SDG-13). Sustainable development means that the exploitation of resources, the direction of investments and technological development must enhance current and future potential for meeting human needs and preserving biodiversity [2]. The Earth's natural environment consists of living species interacting with each other, climate, and weather, affecting humans and their environmental, social, and economic contexts [3]. The built environment provides the setting for human activities, ranging in scale from buildings to cities [4]. The building industry, along with planning authorities, architects, and building clients, is responsible for the development of the built environment and its impact on the natural environment. The construction sector is resource-intensive (using 39% of all energy and 72% of all electricity consumption) and polluting (discharging 40% of all carbon emissions and 30% of all waste output) [5].

To respond to the above challenges, the research aims to explore multidisciplinary (MDC) and transdisciplinary collaboration (TDC) in the nature-based design (NBD) of

the sustainable built environment (SBE). NBD entails learning from nature to develop sustainable solutions, including the integration of nature in buildings [6], multidisciplinary collaboration and approaches for building design and improvement [7–10], innovative building materials [11–13], vegetation on and in buildings [14–16], planning of settlements [17–24] and infrastructures [25], as well as collaboration management [26–29] and related tools [30–38]. Nature-based solutions (NBS) are defined as actions to protect, sustainably manage, and restore natural and modified ecosystems, which address societal challenges (e.g., climate change, food, and water security or natural disasters) effectively and adaptively while supporting human wellbeing and providing biodiversity benefits [39].

Multidisciplinary collaboration (MDC) encompasses the disciplinary interests of its respective members along with providing a comprehensive approach to complex problems [40]. In a multidisciplinary setting, individuals from various disciplines contribute their disciplinary perspectives to solve complex problems that individual disciplines cannot [41]. MDC may evolve into interdisciplinary collaboration (IDC) if there is increased intercommunication [42]. IDC is an activity that exists among existing disciplines or in a reciprocal relationship between them without negating the independence of each discipline [42]. IDC is highly integrated, and methods and perspectives of each discipline are employed to synthesize knowledge and provide insights and solutions to a problem [43].

TDC is completely integrated [43] as it combines more disciplinary contributions from different disciplines to generate a comprehensive level of understanding by creating a systemic framework of several disciplinary and interdisciplinary contributions [44]. TDC admits and confronts complexity in science and challenges knowledge fragmentation as it deals with research problems and organizations that are defined from complex domains such as sustainable development, global environmental change, and public health challenges [45]. Transdisciplinary knowledge is a result of inter-subjectivity as it is a research process that entails the practical reasoning of individuals with the complex nature of social, organizational, and material contexts [44]. TDR and practice need continuous collaboration during all phases of a research project or the implementation of a project [44]. TDC is crucial since it is action-oriented bridging across disciplinary boundaries as well as between theoretical development and professional practice [46]. Transdisciplinary contributions deal with real-world topics and generate knowledge that not only addresses societal problems, but also contributes to the solution [47]. Transdisciplinarity has not been restricted to scientific research but has also been utilized in the practice of architecture, urbanism, and land-use planning that involve stakeholders in decision-making processes [44]. Architecture and urban planning are fertile grounds for transdisciplinary contributions due to their multidisciplinary nature, involving natural and social sciences as well as action-oriented practices aimed to transform the built and natural environments [47]. These circumstances respond to the precondition for TDC: disciplines capable of a constructive dialogue with other domains of knowledge [48]. Due to these reasons, during the last two decades, there have been numerous publications about transdisciplinarity in a wide range of journals, conference proceedings and books with topics related to architecture, urban planning, environmental issues, future studies, public health, and sustainable development [49].

The initial research intention was to identify how TDC could support NBD as it is essential for sustainability [50–52]. NBD methods include biomimicry (or biomimetics), biomorphism, biophilia, bio-utilization, and zoomorphism. Vincent et al. [53] (p. 471) defined biomimicry as "the imitation of the models, systems, and elements of nature for the purpose of solving complex human problems". For many architects, designers, and engineers, biomimicry is a design approach, design principle and part of cross-disciplinary innovation [54,55]. Biomimetics have been used in the development of new materials, modes of operation, design, and management, to improve mobility, physical processes, properties, forms, and structures [56]. Grigson [57] (p. 107) proposed a definition of biomorphism as "the modelling of artistic and engineering design elements based on naturally occurring patterns or shapes present in living organisms". Biomorphism stimulated investigations on natural shapes, materials, structures, and mechanisms [58–60]. It was

prominent in the art of Surrealism and in architecture of Art Nouveau style (1890–1910). The architects Antoni Gaudí, Hector Guimard, and Victor Horta applied biomorphism in their designs. Biophilia is considered as an urge to affiliate with other life forms [61–63]. Kellert [62] believed that biophilia is related to the species' biological development in an adaptive response to the natural world and not to the artificial world created by humans. He proposed a framework for integrating biophilic approaches in architectural and urban design through the direct and indirect experience of nature and the experience of space and place [62]. Hoyos and Fiorentino [64] defined bio-utilization as biotechnologies that use biological resources, providing as an example the use of enzymes in bio-bleaching to reduce the consumption of chlorine-based bleaching agents and minimize the environmental impact of the pulp and paper industry. Zoomorphism is viewed as an "art that portrays one species of animals like another species of animals or art that uses animals as a visual motif" [65] (p. 100). Zoomorphic architecture uses animal morphology as a direct model for architectural design [66].

Considering that the building industry is responsible for the negative impacts of the built environment on the natural environment, the overarching research aim is to develop a transdisciplinary collaboration framework (TCF) that will facilitate TDC for the NBD of the SBE. This need has been identified through a systematic literature review and a case study on MDC practices in the NBD of the SBE in engineering and science fields, which are presented in this paper. The literature review identified various knowledge gaps, which included: (1) how can universities proceed from multidisciplinarity to interdisciplinarity and then to transdisciplinarity; (2) what are the connections between multidisciplinary design and TDR; (3) how can knowledge from multidisciplinary project teams be retrieved and disseminated to ensure that the right level of detail and type of knowledge is gathered during the design process; and (4) how can MDC aided by co-design, participatory design processes, collaboration management, and related tools contribute to support sustainable building design (SBD) of various building types complementing the SBE. The case study was needed as the literature review identified that there has been no research on whether and how TDC had been organized to undertake NBD.

Section 2 describes the above research methods in more detail. The literature review and identified knowledge gaps are presented in Section 3. Section 4 focuses on the case study. A discussion on the research findings is presented in Section 5, as well as the research limitations and potential further research. Section 6 outlines the conclusions of the research.

## 2. Materials and Methods

TDC on the NBD of the SBE in engineering and science fields requires knowledge from different disciplines. The research methods include a systematic literature review on the application of multidisciplinary research (MDR) in building design and a case study on the application of NBD methods in engineering and science. A systematic literature review (SLR) was conducted to identify studies particular to the research topic [67]. The search string "multidisciplinary approaches for sustainable architecture" was used for gathering studies available on ScienceDirect, Google Scholar, Cambridge University Press and Web of Science, leading to the identification of 16,736 studies. The next step was to delete duplicates, followed by screening by keywords from other fields (e.g., medicine, cancer, disease, drug, tissue). The remaining 13,110 studies were scanned by title and abstract, leaving 1560 studies for which inclusion and exclusion criteria were established. Inclusion criteria entailed online accessibility of research papers published in English and related to multidisciplinary approaches for sustainable architecture. Exclusion criteria encompassed research papers from other fields such as business, commerce, and mathematics. A total of 65 studies were included in the literature review.

## 3. Literature Review

Four main research themes were identified in the selected studies: (1) differences between interdisciplinary, multidisciplinary, and transdisciplinary research; (2) challenges

of multidisciplinary design; (3) multidisciplinarity in education and research on building design; and (4) multidisciplinary approaches and collaborative practices for sustainable architecture and urbanism.

### 3.1. Differences between Multidisciplinary, Interdisciplinary, and Transdisciplinary Research

MDR comprises various disciplines working in a self-contained manner; interdisciplinary research (IDR) consists of an issue being approached from a range of disciplinary perspectives; and transdisciplinary research (TDR) involves investigators agreeing that the focus of research is on the organization of knowledge around a complex and diverse domain rather than their disciplines and subjects [68]. Urban studies are considered as a multidisciplinary discipline that brings together experts from both social and natural sciences, including architecture, economics, engineering, geography, human ecology, political science, and sociology [48,68]. MDR is an additive and not an integrative style of research [69], IDR creates knowledge that exceeds the boundaries of a single discipline and integrates methodologies of different disciplines [70], and TDR arises when researchers from different disciplines surpass their separate theoretical and methodological conventions and orientations to develop a common conceptual framework [69]. TDR is not a "new discipline" but a manner of seeing the world holistically [71].

Uni-disciplinary research concerns a single discipline and one level of reality, whereas TDR concerns several disciplines and various levels of reality. Considering that higher-education institutions (HEIs) reinforce "uni-disciplinary" formation at the undergraduate level, it has been suggested that the steps for transformation towards MDR should be oriented at the postgraduate level through thematic areas beyond specific disciplines [71]. A study that analyzed 90,000 publications of MDR findings by 2500 researchers in over 100 universities identified that interdisciplinarity is generated through research multidisciplinarity, which is achievable through collaborations occurring within multidisciplinary institutions between academics with different research interests [72].

### 3.2. Challenges of Multidisciplinary Design

Although architects aspire to coordinate multidisciplinary design, they face challenges in achieving multidisciplinarity due to a potential lack of professional development programs ensuring the appropriate training to apply environmentally friendly principles to all project types [73]. Individual team members may not be prepared to promote their potential contributions and may not know how to seek, identify, value, or use either information from clients or expertise from other team members [74]. Multidisciplinary design faces communication challenges appearing beyond cultural differences in diverse teams where there may be a lack of understanding due to language variation or jargon idiosyncrasy [75].

Both strong disciplinary knowledge and excellent communication skills are needed for multidisciplinarity [76]. Multidisciplinarity requires good teamwork that needs sufficient common ground among disciplines to support the sharing of knowledge and subsequent resolution of a shared problem [77]. Researchers and practitioners often structure their collaboration using ICT tools for better teamwork, problem definition, and resolution [77]. Conflicting data-collection requirements are a potential disadvantage of multidisciplinarity as they hinder the project team's ability to meet all their members' objectives [78]. These challenges can be mitigated by establishing a systematic process through which a multidisciplinary project team can agree to a primary goal, communicate effectively, improve their ability to make the right decisions on necessary trade-offs, and balance project objectives with available resources [78].

### 3.3. Multidisciplinarity in Education and Research in Building Design

AEC students need to learn how to exchange information and work in multidisciplinary teams [79], including multiple aspects such as energy efficiency, the possibility of designing buildings at nearly zero energy consumption, the sustainable use of renewable and non-renewable resources, and the use of recycled building products [80]. The examples

below illustrate specific approaches and processes that contribute to multidisciplinarity in education and research in building design.

The University of Illinois and University of Florida, USA, designed the master's course on Collaborative Design Processes (CDP) by using information communication technology (ICT) methods for collaborative design in the AEC industry and enabled multidisciplinary student teams (structural engineering, architecture, and construction management) to collaborate from remote locations via the Internet [81]. The students made recommendations for improving software tools for collaborative multidisciplinary design [81].

The knowledge-capture report (KCR), developed at Loughborough University, UK, aimed to capture the created and utilized knowledge of AEC projects to improve the quality and effectiveness of future projects [82]. The participants involved in the case study agreed that the KCR enabled multidisciplinary teamwork but did not allow capturing precise knowledge that could successfully integrate various engineering fields [82].

MDR of academics and scientists from architecture, climatology, and psychology at the University of Gothenburg and University of Gävle, Sweden, investigated urban public spaces and their microclimates in Gothenburg, Sweden. They analyzed the impact of three weather variables (clearness index, air temperature, wind) on participants' perception of the current weather and on their behavioral, aesthetical, and emotional assessments of four urban public spaces (square, courtyard, park, waterfront) [83].

Academics at the Civil Engineering Faculty of the Technical University of Kosice, Slovakia, founded the Institute of Building and Environmental Engineering offering bachelor's, master's, and doctoral programs that focus on interdisciplinary areas of building design [84]. The programs have led to an increase in interest in multidisciplinary education among civil and HVAC engineers, environmental medicine doctors, and building inspectors [84].

"Integral Design", which is focused on sustainable climatic design, was developed at the Faculty of Architecture, Building, and Planning at the Eindhoven University of Technology, Netherlands. This master's program is offered to students of architecture, building physics, building technology, building services, and structural engineering. The workshops have become a part of the professional educational program of Royal Institute of Dutch Architects since 2006 [85]. Professionals usually value new design methods lower than students [27,85,86], but in this case the practitioners valued the proposed design method higher than the students [87].

A team of five students from architecture, building technology, electrical engineering, structural engineering, and building services from the Czech Technical University of Prague applied the Integral Design approach [86] in their competition entry for the US Department competition Energy Solar Decathlon 2013 [88].

Academics and researchers at the School of Planning, Design and Construction, Michigan State University, USA, designed an undergraduate course "Special Topics in Planning, Design and Construction", focused on teaching sustainability to students of construction management, interior design, urban planning, and landscape architecture, generating a higher level of interaction to produce more creative solutions for better-articulated projects [79].

The Department of Civil, Environmental and Architectural Engineering, University of Padua, Italy, developed a methodological pathway for innovative sustainable design that acknowledges the importance of environmental context and the significance of the application of sustainability assessment criteria in relation to project requirements. The methodology has facilitated the analysis process for innovative solutions for buildings at nearly zero energy consumption [80].

Academics from the University of Lausanne and Swiss Federal Institute of Technology (EPFL), Lausanne, Switzerland, were inspired by the landing of Curiosity on Mars in 2012 and set up a Learning Unit "Building on planet Mars" at the School of Architecture, Civil and Environmental Engineering at EPFL that focused on special building projects for space

systems by offering seminars on the planet and human needs that informed students' designs for a self-sufficient Martian base for 30 people [89,90].

New trends in green building design and BIM transforming the AEC industry were explored through a project at California State University, Fresno, CA, USA [91]. The pilot study provided evidence of the efficiency of project-based learning through a joint course as a pedagogical approach to instill in students the ideas such as the green building design process with BIM facilitation stimulated in a multidisciplinary project environment [91]. The post-design survey highlighted an increase in the perception and ability of 3rd-semester students in problem solving for subjects such as biology [92].

Green Regeneration, Environmental, Energy Update at the School of Engineering, University of KwaZulu-Natal, Durban, South Africa, was a multidisciplinary program that interlinked research, teaching and learning, energy management, and sustainability upgrade initiatives of the university's buildings and facilities. The research results indicated the effectiveness of an innovative multidisciplinary program that promotes sustainable development through higher education [93]. As successful innovation requires collaboration between experts in engineering, design, and business, curricula should include knowledge in business, impact measurement, systems thinking, communication, and social justice [94].

As the built environment is one of the leading contributors to climate change, resource depletion, waste generation, pollution, poverty, and inequity, the ISAlab workshops at Universitat Politecnica de Valencia, Spain educated future engineers to find robust ways to implement sustainability at a practical level by taking an account of multidisciplinary perspectives regarding the solutions to real-life problems [95]. The research identified several interpersonal characteristics of individuals who work well in a transdisciplinary environment, including complex and interlinked thinking, open-mindedness and empathy, and the ability to reflect on knowledge integration [95]. The development of multidisciplinary approaches for AEC students for SBD also relies on the acceptance of the unknown, the flexibility of thought and purpose, a common goal, and institutional support of a shared programmatic and course-level vision as well as the provision for iterations and risk-taking in multidisciplinary curricula [96].

### 3.3.1. Generalized Views and Conclusions

Some of the studies from this theme identified that ICT methods [81], the Integral Design method [33,86–88], project-based learning [89–91], and design charettes [92] are amongst the most practiced approaches and processes used by academics and researchers for collaborative design in the AEC industry to enable multidisciplinarity in education and research in building design. The Integral Design method developed by Zeiler et al. [85] facilitated multidisciplinarity as it formed a bridge between building industry professionals and engineering students. This method was tested by other researchers [88] for design competition entries to pursue sustainable solutions. Project-based learning [97] also facilitated multidisciplinary education because it enhanced academic achievement, learning permanence, and learning functionality. Further research expanded this approach as a few pilot studies creatively utilized project-based learning through joint courses to create multidisciplinarity environments [91].

Studies from this theme also outlined several barriers and limitations that students and instructors faced in their joint agenda to enhance multidisciplinarity in education and research in building design. Some of these barriers related to multidisciplinary project teams included the lack of knowledge of students about other disciplines, the lack of integrative knowledge and abilities in a project team, and the variation in cultural expectations based on individuals and disciplines. Other researchers also suggested that the testing of the effectiveness of multidisciplinary student teams requires a larger number of students in order to lay the foundation of a future statistical analysis of teamwork in SBE projects and teaching at HEIs. As a formative and summative observation during a multidisciplinary project limits the amount of data collected [98,99], more observation sessions are essential.

### 3.4. Multidisciplinary Approaches and Collaborative Practices for Sustainable Architecture and Urbanism

Evidence of MDC has been provided in numerous studies related to building design and improvement, building materials, vegetation on and in buildings, the planning of settlements and infrastructures, as well as in collaboration management and related tools.

#### 3.4.1. Building Design and Improvement

Multidisciplinary approaches are applied in the design of building envelopes as interfaces between the outdoor environment and indoor occupied spaces [7]. Nature provides multiple examples of adaptation strategies that could be applied in their design [8]. Biomimetics is an emerging multidisciplinary design field in architecture that allows designers to tackle difficulties throughout the design process, especially where biophysical information is not easily accessible. The interdisciplinarity of biomimetics entails the design concept generation through an exploration of three different domains: problem, nature, and solution, where collaborations between scientists from diverse disciplines support the transformation of concepts from nature into technical solutions [7].

Thermal regulation used by some cold-blooded animals following changes in temperature inspired new solutions in the design of building energy systems [8]. Different animal thermoregulation strategies were defined (biological domain), examined and classified into three categories: those corresponding to a system, a process, or an action. This classification is essential to formulate the parallelism with building systems (transfer phase). Based on the animal thermoregulation strategies and identification of similar functions in buildings, several technological hypotheses (technological domain) were proposed and classified into three groups: building, systems, components [8].

Heritage conservation and renovation require complex multidisciplinarity to integrate knowledge from several disciplines, enabling the adoption of well-balanced solutions with respect to heritage values, encompassing a building survey, monitoring campaign, on-site investigation, and energy modeling [9]. The proposed methodology was tested on Giuseppe Terrangi's Casa del Fascio in Como, Italy, leading to the modern reinvention of internal curtains.

Multidisciplinary approaches that include collaboration with prison officers, management, and staff, as well as prisoners were utilized to redesign prisons to be socially sustainable and to reappraise the ethics of justice that reconsiders the status of people on custodial sentences, resulting in the development of the design guide on "Building a Rehabilitative Culture and Wellbeing in Prison Design" [10].

#### 3.4.2. Building Materials

Takano et al. [11] demonstrated the significance of building material selection using multidisciplinary parameters (environmental and economic) on a three-story townhouse planned in Helsinki, Finland. The effects of material choice for three building component categories were presented and compared: structural frame, inner components (insulation and sheathing), and surface components (exterior cladding and flooring). Environmental parameters were more impactful compared to economic parameters as they vary more between building materials [11]. They believe that biomimetic structural materials can meet the strict requirements of engineering materials [11].

Zhang et al. [12] endorsed the research of Takano et al. [11] by suggesting that multidisciplinary scientists have a profound interest in natural material characteristics such as structure, composition, and interface. Living organisms are an inspiration for manufacturing artificial materials because they utilize limited elements to control local properties for adapting to the specific environmental requirements [11]. Structural material selection was identified as the most influential as it forms the building envelope, which is a critical building element [11,13].

### 3.4.3. Vegetation on and in Buildings

The use of vegetation in architecture is a design strategy aiming to improve building performance and appearance, but also the outdoor climate [14]. A multidisciplinary approach and various skills are needed from the early design stages to optimize all aspects [14]. Various technological solutions have been proposed over the years to cover buildings with vegetation, such as green roofs, green walls, and green balconies. The vertical greenery modular system (VGMS) is a particular typology of green wall that has gained high consensus among designers due to the positive impacts of this technology relating to façade orientation, building functionality, climatic conditions, plant typology, wall assemblies, and mechanical and technological issues [15]. The VGMS was tested by the Green Envelope System research project in Turin, Italy. The VGMS provides multiple services in an urban context due to its interdisciplinary and multiscale approach, allowing designers to efficiently combine different materials/species/technical solutions according to the project goals and expected results (aesthetic value, saving energy, noise reduction, cost reduction, etc.). Increasing the number of choices of sustainable materials and reducing the energy demand during winter would enable an increase in vertical greenery systems integrating $CO_2$ absorption, biodiversity, run-off, etc. [16].

### 3.4.4. Planning of Settlements

The investigation of cities requires the collaboration of numerous disciplines through MDR and IDR, including several interdisciplinary bridges for knowledge transfer. Two approaches to studying cities are observed: (1) the city and its development subjected to the laws of nature, and (2) the cultural vision of urban development, confronting different schools of thought related to the nature/culture dichotomy [17].

An example of MDR on cities is the project on Dispersion of Air Pollution and its Penetration into the Local Environment (DAPPLE) that brought together an MDR group from six UK universities and undertook field measurements, wind tunnel modeling, and computer simulations to understand physical processes affecting the street- and neighborhood-scale flow of air, traffic, and people, and their corresponding interactions with the dispersion of pollutants at street–canyon intersections [18]. The DAPPLE consortium collected data for turbulent winds, curbside carbon monoxide and nitrogen dioxide pollution levels, traffic flow and composition, and pollution levels within the traffic flow along Marylebone Road, London.

James et al. [19] presented an example of a research framework for enhancing MDR on urban green space, conceived by using a Delphi technique consisting of email-mediated discussions and a two-day symposium involving experts from various disciplines and from several European countries. The discussions formulated 35 questions categorized into five themes: physicality, experience, valuation, management, and governing of urban green space, providing a framework for urban green space research [19].

Multidisciplinary approaches allow the integration of community structure, local traditions, cultural diversity, and disaster management planning to design possible models and frameworks for the construction of sustainable new dwellings/communities in earthquake zones [20]. Various hazard-prone regions in Romania were studied to understand the architecture for emergency events relating to requirements such as: (1) responsiveness to the local and specific needs; (2) the mapping of migration paths; (3) designing with the involvement of the community; (4) fighting mental alienation; (5) finding independent funds from United Nations or other transnational organizations to supplement those provided by local authorities [20].

A multidisciplinary methodology was presented that covered various disciplines such as architecture, computer science, civil engineering, electrical, electronic and telecommunication engineering, social science, and behavioral science to successfully implement the development of suitable modeling tools and real solutions of sociotechnical systems [21]. ICTs and smart cities promise a more efficient use of physical infrastructure and resources, adaptability to changing circumstances, and effective engagement with

citizens. The research highlights the pivotal role of the multidisciplinary/multi-partner approach, illustrating how different applications are instantiated by various pilot projects within several major European Union (EU) and national research programs, and the Bologna municipality in Italy [21].

Castelli et al. [22] proposed the Urban Intelligence (UI) framework as an ecosystem of technologies to improve the urban environment, wellbeing, quality of life, and smart city systems [22]. The UI develops a digital twin of a city through a cyber-physical space of all city systems and subsystems. The main characteristics of the UI framework are: (1) the complete multidisciplinary integration of city layers handled by multidisciplinary analysis schemes, (2) the multidisciplinary model-based optimization of multiple city layers, (3) the integration of "human-oriented" information by participative strategies, and (4) the modularity of the application. The UI architecture contributes to solving problems and making high-level decisions due to coordinated multidisciplinarity facilitating the unification of digital models of cities.

MDR related to small and medium towns (SMTs) need better integration related to urban dimensions within a broader geographic approach [23]. Five levels of multidisciplinary approaches were addressed to pinpoint the theoretical grounds for promotion and advocacy of SMTs as major drivers of regional sustainability: (1) agglomeration advantages and networking efficiencies through the strict accounting of costs and benefits; (2) clustering online environments and their extension to open networking systems; (3) sustainable innovation for SMTs, technology, and knowledge transfer in open innovation systems—as settings for discussions framing new technological developments and application of artificial intelligence; (4) knowledge and new technological developments with local spill-overs enhanced by new educational programs and learning diffusion at advanced levels; and (5) social functions of SMTs addressed in the disciplines of sociology, architecture, and planning [23].

Kopp et al. [24] outlined the need for integration of blue–green infrastructure (BGI) to support the multidisciplinary planning of sustainable cities. They identified six approaches to BGI: stormwater management, hydrological processes, ecosystem services, land cover, green networks, and water corridors. The typology of BGI approaches indicates that the design and planning using BGI needs multidisciplinary and integrated concepts that are consistent with SDGs such as SDG 6 (Clean Water and Sanitation), SDG 9 (Industry, Innovation, and Infrastructure), SDG 11 (Sustainable Cities and Communities), SDG 13 (Climate Action), and SDG 15 (Life on Land) [24].

### 3.4.5. Infrastructures

Multidisciplinarity is necessary to develop a mutual understanding between modeling team members and enhancing quality assurance in model-based water management [25]. The investigation indicated that enhanced quality assurance helped avoid frequent shortcomings such as malpractice and a tendency to oversell model capabilities. A software product supported the full modeling process through an ontological knowledge base and a modeling support tool (MoST) managing complex processes and guiding users [25]. The advantages of the MoST include guidance on the best multidisciplinary modeling practice and continuous monitoring of the modeling team (modeler, water manager, auditor, stakeholder, concerned members of the public).

### 3.4.6. Collaboration Management

Architecture as a discipline thrives on a design team's creative synthesis if supported by flexible generic frameworks that sustain group dynamism through a project's lifetime [26]. Multidisciplinary or interdisciplinary collaboration helps avoid a building design to be divorced from legal, technical, political, and economic contexts [26]. The collaborative design framework of the Pompidou Centre in Paris, France, consisted of three levels: (1) group process design integration, (2) interplay between integration and distribution,

and (3) design distribution. The challenge of developing a responsive multidisciplinary architectural design team lies between the design integration and distribution processes [26].

Zeiler and Savanovic [27] proposed a comprehensive approach to collaborative design management integrating design and engineering knowledge in the conceptual stage of building design. The research implemented the concept–knowledge theory to increase knowledge creation and stimulate knowledge exchange within the building design team [27]. The theoretical approaches were tested through several workshops involving diverse professionals assigned to design teams, which ideally included an architect, a building physics consultant, a building services consultant, and a structural engineer. The workshops' findings indicated that architects should improve collaborative design and be multidisciplinary leaders as they are the first actors in the design process. Leon et al. [28] also believe that the effective MDC of stakeholders during the feasibility and concept stages of a project provide the highest potential for a successful building project. Their research mapped design processes within different disciplines and information management across early design stages to establish a conceptual design protocol for testing and comparing the design paradigms [28]. Bonenberg and Kapliński [29] presented an architect's attitudes towards sustainable development. Both studies [27,29] highlight the role of an architect in implementing multidimensional processes of sustainable design, suggesting that an architect must be able to design energy-efficient buildings and avoid the loss of prestige, technical mistakes, and conflicts with participants and stakeholders. Two types of knowledge are needed for supporting this transition: formal (explicit knowledge) and hidden (tacit knowledge) [29].

### 3.4.7. Collaboration Tools

The advancement of information and web-based communication technologies supports architecture, engineering, construction, and facilities management through collaborative creation, management, dissemination, and information usage throughout the project lifecycle and by integrating people, processes, and business systems more effectively [30].

Goel et al. [31] developed a knowledge-based CAD system supporting biologically inspired design, named Design by Analogy to Nature Engine, that consists of four dimensions: cognition, collaboration, conceptual design and creativity, and provides functional models of biological systems to help designers conceptualize complex designs.

Multidisciplinary design optimization (MDO) emerged in the AEC industry, assisting designers to make efficient decisions and achieve design alternatives faster. Five tools used for MDO in the AEC industry were tested, which included optimization platforms, software tools, building information modeling (BIM), parametric modeling (PM), and analysis tools, such as structural analysis, energy simulation, cost analysis, etc. [32]. The research explains the challenges and strategies to develop MDO [32]. Some of the strategies included testing the tool interfaces to validate computational processes by running automatically and testing if parametric relationships and components were possible. Technological, documentation, and programming challenges emerged when testing tools for MDO in the AEC industry. Some of the technical requisites for MDO include component interoperability, tool automation, and model parametrization capabilities [32]. The research results help researchers develop MDO by integrating appropriately suitable computer applications to create a design and optimization workflow.

Almeida et al. [33] proposed an Adaptive Multi-platform Multilingual Multidevice Multimodal Interaction (AM4I) framework and demonstrated its potential by a proof-of-concept application in different smart environment contexts [33]. Frameworks such as AM4I need multidisciplinary user-centered tools to understand the living patterns of people in smart environments. Multidisciplinary insight for smart environments is enhanced by supporting iterative design, system development, and prototyping evaluation cycles [33].

Parametric tools have a considerable impact at the concept stage [27–29] on developing design awareness regarding environmental issues in building design and urban planning for providing initial interdisciplinary solutions [34]. The study identifies Antoni Gaudí,

Frei Otto, and Sergio Musmeci as pioneers of the use of gravity modeling as the single independent parameter in the analysis of structural shapes. Their analysis included algorithmic models that dealt with building form optimization, urban scale, sun radiation levels, shadow, and wind analyses, indicating the use of solar and wind energy as alternative energy sources [34].

Çetin et al. [35] investigated digital technologies that could enable the circular economy in the built environment. Their study identified that BIM is an essential tool throughout the entire lifespan of buildings as it offers effective information sharing and transparent project coordination [36–38]; however, in practice it is not mature for all life cycle stages [35]. The use of digital twins offers an integrative platform on which various technologies are combined to represent the real world at the building, portfolio, and urban levels, as well as enables the monitoring and management of resource flows of the built environment [35].

The examples of multidisciplinary approaches from the seven themes (building design and improvement, building materials, vegetation on and in buildings, the planning of settlements and infrastructures, as well as in collaboration management and related tools) show where multidisciplinarity has taken place.

Table 1 highlights the knowledge gaps identified from the systematic literature review. The overview of the seven perspectives shows that MDR is taking place in different areas of engineering, but nature-based design framework rarely entails MDC. The next section presents a case study as the literature review identified that there has been no research on whether and how MDC/IDC/TDC had been organized to undertake NBD.

**Table 1.** Identified knowledge gaps.

| Literature Review Themes | Knowledge Gaps |
|---|---|
| 1. Differences between multidisciplinary, interdisciplinary, and transdisciplinary research | How can universities proceed from multidisciplinarity to interdisciplinarity and then to transdisciplinarity? |
| 2. Challenges of multidisciplinary design | What are the connections between multidisciplinary design and TDR? |
| 3. Multidisciplinarity in education and research in building design | How can knowledge from multidisciplinary project teams be retrieved and disseminated to ensure the right level of detail and type of knowledge is gathered during the design process? |
| 4. Multidisciplinary approaches and collaborative practices for sustainable architecture and urbanism | How can MDC aided by co-design, participatory design processes, collaboration management, and related tools contribute to support SBD of various building types complementing the SBE? |

## 4. Case Study

### 4.1. Introduction

As the reviewed literature did not identify how MDC/IDC/TDC is organized for NBD as perceived by academics, a case study was developed to overcome this knowledge gap. The case study explored multidisciplinary approaches for NBD in engineering and science fields.

The case study aimed to: (1) identify NBD related research in the Faculty of Engineering (FoE) and the Faculty of Science (FoS) at the University of Strathclyde (UoS) and investigate if they could be applicable in building design; (2) explore the extent and type of collaboration between different disciplines in researching the potential application of NBD in general, not only in architecture. An online survey was distributed and online interviews with academics were undertaken.

Access to the Pure system at the University of Strathclyde enabled the easy identification of research outputs related to NBD and their authors for this case study. The search in the Pure system [100] through 6889 research outputs of academics in both faculties

identified 60 outputs relating to NBD, produced by 75 academics who were principal investigators or co-investigators.

Nine overarching themes and 26 subtopics were identified in the selected research outputs (Table 2). These themes included inspiration, modeling and management, manufacture, design, materials, reuse, carbon dioxide reduction, research methods, and ecological living. All data were anonymized, and the academics were numerically listed (A1–A75).

**Table 2.** Overarching themes and sub-topics of identified research outputs by academics of FoE an FoS at UoS.

| Theme | No. | Topic | Academic |
|---|---|---|---|
| | 1.1 | Biophilia | 1 |
| | 1.2 | Biological approaches | 61, 62, 63, 72, 73 |
| 1. Inspiration | 1.3 | Biomimetics | 56, 57 |
| | 1.4 | Bio-inspired movements | 31 |
| | 1.5 | Bio-inspired systems | 5, 6 |
| | 2.1 | Ecosystem modeling | 65 |
| 2. Modeling and management | 2.2 | Water management | 16, 25 |
| | 2.3 | Pollution | 38 |
| | 3.1 | Bio-processibility | 8 |
| 3. Manufacture | 3.2 | Bio-fabrication | 3 |
| | 3.3 | Biomanufacturing | 74, 75 |
| | 4.1 | Design optimization | 58, 59, 60, 71 |
| 4. Design | 4.2 | Intelligent design | 24 |
| | 4.3 | Functional surfaces | 9, 10 |
| | 5.1 | Plant-based materials | 4, 70 |
| 5. Materials | 5.2 | Sustainable materials | 14, 15, 30 |
| | 5.3 | Nanomaterials | 7, 11, 53, 54 |
| 6. Reuse | 6.1 | Circular economy | 2, 33, 34, 44, 50, 5 |
| | 6.2 | End-of-life strategy | 32 |
| 7. CO$_2$ reduction | 7.1 | Energy management | 45, 46, 47, 48, 49, 52, 55 |
| | 7.2 | Low-carbon technologies | 39, 40, 41, 42, 43, 64 |
| 8. Research methods | 8.1 | Multidisciplinary research | 12, 13, 17, 18, 19, 20, 21, 22, 23 |
| | 8.2 | Interdisciplinary research | 66, 67, 68, 69 |
| 9. Ecological living | 9.1 | Smart planning strategies | 28, 29 |
| | 9.2 | Future cities | 35, 36, 37 |
| | 9.3 | Rural development | 26, 27 |

Based on their research interests and the description of their projects, 75 bespoke introductions and questionnaires were prepared. The respondents had the option to return the completed questionnaire or discuss the answers in an online interview. A total of 13 positive responses (Table 3) were received with at least one response per theme except for the themes of modeling and management, and manufacture.

**Table 3.** Themes, positive responses of academics and their faculties.

| No. | Themes | Positive Responses | Faculty |
|---|---|---|---|
| 1. | Inspiration | A6, A72 | Engineering, Science |
| 2. | Design | A10 | Science |
| 3. | Materials | A11, A54, A70 | Engineering, Science, Science |
| 4. | Reuse | A2 | Engineering |
| 5. | CO$_2$ reduction | A42, A45 | Business, Engineering |
| 6. | Research methods | A12, A13 | Engineering, Engineering |
| 7. | Ecological living | A27, A35 | Humanities, Engineering |
| Total | | 13 | |

Appendix A contains a summary of the information provided by the academics in the questionnaires and interviews.

*4.2. Survey Findings*

This section presents the most significant research findings (Table 4) highlighting several topics such as the benefits of NBD, the scale of the applications of NBD approaches, the challenges for applying NBD in the built environment, and overcoming the barriers to the development and application of NBD approaches in the built environment.

**Table 4.** Most significant survey findings.

| Survey Findings | Reasons for Consideration |
|---|---|
| 1. Benefits of NBD | 1.1. To move away from fossil fuels<br>1.2. Carbon dioxide reduction<br>1.3. Biodegradable or compostable materials<br>1.4. Reusing waste within the circular economy<br>1.5. To respond to complex, highly interdependent urban systems |
| 2. Scale of applications of NBD approaches | 2.1. Urban systems<br>2.2. Rural areas<br>2.3. Infrastructure: accessibility and affordability of energy from renewable sources; smart systems in public transport; waste recycling; vertical farming; minimizing environmental impacts; improving the quality of life of citizens.<br>2.4. Building elements: structural integrity; building insulation<br>2.5. Building materials: surface coatings to improve building performance; self-cleaning materials; reinforcing materials; building complex structures; developing nanostructures |
| 3. Challenges of applying NBD in the built environment | 3.1. Engagement of stakeholders and researchers<br>3.1.1. Lack of engagement with beneficiaries<br>3.1.2. Low awareness, planning restrictions, and legal and financial barriers<br>3.1.3. Establishing a co-development environment comprising relevant experts, existing service providers, and citizens to develop a solution and build a "sense of ownership"<br>3.2. Lack of knowledge<br>3.2.1. Knowledge of working mechanisms in the biological world<br>3.2.2. Lack of understanding of nature's multiparameter space, its mechanisms, and the interrelation of its counterparts<br>3.2.3. Complexity<br>3.3. Practical concerns<br>3.3.1. Focus on functionality and financial costs<br>3.3.2. High energy lab and material costs, and wasteful solvent disposal<br>3.3.3. The need for and cost of testing the performance of reused building elements and materials<br>3.3.4. Finding professionals who are specialists in NBD<br>3.4. Research challenges<br>3.4.1. Ability to control molecules at the nanoscale and extrapolate that to an architectural level<br>3.4.2. Ability to convert materials back to raw materials<br>3.4.3. Uncertainty whether a sustainable material will have enhanced functionality; nanocomposite materials could be environmentally friendly but might not be completely sustainable as scientists aim for functionality that could conflict with sustainability<br>3.4.4. Using a natural or sustainable material does not ensure carbon emissions reduction as processing these materials can be more energy-intensive and emits more carbon unless there is a sustainable energy source<br>3.4.5. Creating high-integrity modeling and monitoring tools requiring the development and validation of detailed algorithms for different aspects such as heat transfer, fluid flow, weather, and human behavior<br>3.4.6. Extensive and complicated range of scale among the disciplines of chemistry, biology, and engineering for MDR as chemists use nanometers, biologists use micrometers, and engineers use millimeters and centimeters<br>3.4.7. Professionals never fully understand each other, e.g., a chemist would understand finite element analysis but not its engineering aspects<br>3.5. Educational challenges<br>3.5.1. Departments are reluctant to implement MDC due to issues such as credit weighting, staff workload allocation, stakeholder management, difficult coordination, disagreement on ownership, and unbalanced sharing of responsibility |

**Table 4.** *Cont.*

| Survey Findings | Reasons for Consideration |
|---|---|
| 4. Overcoming the barriers to the development and application of NBD approaches in the built environment | 4.1. Policies and strategies<br>4.1.1. There needs to be both ecological and social commitment to reduce carbon emissions and reduce inequality<br>4.1.2. Circular economy encourages reusing natural resources as it mimics natural processes that do not produce waste<br>4.1.3. Consider local communities when identifying available natural resources to achieve an optimum solution<br>4.1.4. Policies are important to support the design of sustainable architecture, e.g., subsidies to young families living in rural areas in exchange for complying with criteria for sustainable building design<br>4.1.5. Bio-based strategies improve rural lifestyle as they encourage using local resources<br>4.1.6. Self-building invites NBD approach in residential areas<br>4.2. Education<br>4.2.1. Need for formal education in biology and zoology<br>4.2.2. Biologists need to comprehensively enhance existing biological knowledge so architects, chemists, physicists, and engineers can use it<br>4.2.3. Multidisciplinary education of architects can contribute to the design of sustainable architecture<br>4.2.4. Correlation between NBD and carbon emission reduction needs to be taught, using multidisciplinary education<br>4.3. Research strategies<br>4.3.1. Comprehensively examine building materials to achieve their whole lifecycle sustainability<br>4.3.2. Energy demands of manufacturing natural polymers should be lower than fossil-fuel-based plastic production<br>4.3.3. Sustainably sourced nanocomposite materials<br>4.3.4. Effective design decisions need timely information<br>4.3.5. Modeling/monitoring tools need to be integrous to exhibit reality and developed by using theories and methods that are good representations of thermo-fluid mechanisms governing all natural systems. |

The scale of applications of NBD approaches included urban systems, rural areas, infrastructure, building elements, and building materials. Another significant finding comprised challenges of applying NBD in the built environment such as the engagement of stakeholders and researchers, lack of knowledge, practical concerns, research, and educational challenges.

The survey findings also entailed methods to overcome the barriers to the development and application of NBD approaches in the built environment through policies and strategies, education, and research strategies. The results of the survey and interviews with the identified academics revealed problems in organizing TDC for NBD that will inform the development of a TDC framework for NBD.

### 4.3. NBD Applications in Other Disciplines Informing SBD

The application of NBD in other disciplines informs SBD in numerous aspects. The identified applications of NBD contribute to SBD due to reduced carbon emissions from buildings and utilization of waste within circular economy by the AEC industry. The application of NBD approaches in SBD includes accessible and affordable energy from renewable sources that could enable an increase in smart homes and sustainable communities. Surface coatings and self-cleaning materials are applicable to existing and future buildings to improve building performance, contributing to the SBE.

The challenges of applying NBD in the built environment such as the engagement of stakeholders and researchers, lack of knowledge, practical concerns, research, and educational challenges could be subdued through policies and strategies, education, and research strategies. The consideration of local communities when identifying available natural resources contributes to identifying optimum solutions corresponding to the environmental and social sustainability of the built environment. The provision of subsidies to young families living in rural areas increases awareness, reduces financial barriers, and introduces NBD approaches to residential areas, complying with the criteria

for SBD. An increased awareness of NBD could also be enabled through the multidisciplinary/interdisciplinary/transdisciplinary education of future architects and designers to understand the correlation between NBD and carbon emissions reduction.

## 5. Discussion

Each sub-theme related to multidisciplinary approaches and collaborative practices for sustainable architecture and urbanism of the literature review presented various ideas that could be expanded for potential further research. In relation to building design and improvement, it is anticipated that designers would need to consider the underlying environmental processes of numerous distinct morphologies at the initial stages of a design process for the development of adaptive solutions for building envelopes [7]. Designers and engineers need to collaborate to combine parametric software with energy modeling specifically for historic buildings [9]. MDC aided by co-design processes contributes to an improved understanding of the fundamental relationship between people and the built environment [10]; however, more research is needed for the better design of different building types, especially hospitals, education facilities, and shared housing.

Regarding building materials, further studies are necessary to assess bio-inspired structural materials and their role in engineering fields such as machinery, automobile, aerospace, rail transit, and construction services [12]. Vegetation on and in buildings showed that the VGMS allows designers to combine different materials/species/technical solutions according to project goals and expected results; however, additional research is needed on water run-off and its relationship with the loss of water and nutrients in the VGMS [14].

Research about settlements showed that studying cities requires the collaboration of numerous disciplines, but that transdisciplinarity raises the problems of methodology [17]. Further research is necessary for unifying methodologies in order to identify the theoretical points of different disciplines and realities. The research also highlighted that ICTs and smart cities promise an efficient use of infrastructure and resources, adaptability to changing conditions, and effective engagement with citizens [21]. Further research is required to test the implementation of smart cities by various pilot projects in numerous regions facing different climatic conditions.

The UI framework proposed the development of a digital twin organized into layers to improve the urban planning of settlements [22] and presented several areas that need potential further research. These areas include the use of the UI at the strategic level of planning and intervention on a city's infrastructure, systems, and energy distribution; the operational level of the urban planning connected with services management, local mobility planning, building design, and site planning; and for the urgent integration of resiliency and sustainability for emergencies.

MDR related to SMTs presented evidence on how to better integrate urban dimensions within a broader geographic approach through morphological, administrative, and functional perspectives [23]. However, additional studies are needed to consider SMTs' development for territorial sustainability; to test the adaptability and efficiency of small towns to mitigate environmental stressors due to increasing urban sprawl; and to discuss processual and operational innovation of small towns in three distinct functional areas: connectivity, growth, technology.

Regarding infrastructure, it is established that a multidisciplinarity modeling practice can manage complex processes relating to model-based water management using the modeling support tool (MoST) and knowledge base (KB) [25]. Further studies could assess other methods to reduce malpractice (careless handling of input data, inadequate model setup, insufficient calibration and validation, and using models out of scope) in model-based water management.

The subtheme of collaboration management showed that multidisciplinary or IDC during the feasibility and concept stages of a project helps avoid a building design to be divorced from legal, technical, political, and economic contexts [26,28]. Additional

research is needed to understand the effectiveness of a pre-defined design protocol and its relationship with different working environments and social aspects of collaboration of multidisciplinary team members.

In relation to collaboration tools, it is evidenced that information and web-based communication technologies and MDO support the AEC and FM industry and assist designers to make efficient decisions [30,32]. Interoperability of data modeling and integration, and of communication protocols and languages requires further research. Further studies are needed to discover the functionality of MDO in Process Integration and Design Optimization (PIDO) platforms and to explore the usefulness of PM tools to analyze the design space from the conceptual design to the current limitations of MDO.

The research presented evidence that the AM4I framework has the potential to function in different smart environments and open several routes for future work [33]. The interaction design for the SmartHome ecosystem requires further studies on the evaluation of user experience in smart environment contexts, which is being addressed by a partnership between Smart Green Homes and BOSCH [33]. Further research is needed to design and deploy interaction to contemplate how non-domestic buildings such as universities can be integrated with technologies and services to improve their occupants' daily routine.

The case study on the application of NBD in the FoE and the FoS at the UoS focused on identifying the previous research related to NBD with the aim to identify multidisciplinary approaches and solutions applicable to the design of the SBE.

## 6. Conclusions

The systematic literature review and the survey results with academics on MDR collaboration showed a lack of TDR due to limited communication between disciplines, as well as with AEC industry and end users. Therefore, a TCF on research for NBD is crucial to support collaborative research, knowledge transfer and engagement within academia, and with the industry and end users. The evidence from the systematic literature review and the case study of NBD applications in engineering and science will contribute to the development of a general framework for conducting transdisciplinary research. Further research is needed to systematically explore TCFs that address knowledge transfer within academia, engagement with industry, and NBD and NBS for the sustainability of the built environment. Findings from the current and future research will develop and test a general TCF and then to develop TCF for NBD of SBE aligned with the RIBA Plan of Work.

**Author Contributions:** This paper has been written by A.N.B. with supervision by B.D. The co-authors contributed to the paper as follows: Conceptualization, formal analysis, investigation, resources, data curation, methodology, and writing—original draft preparation: A.N.B.; Validation, supervision, project administration, funding acquisition, and writing—review and editing: B.D. All authors have read and agreed to the published version of the manuscript.

**Funding:** PhD scholarship was provided by the University of Strathclyde. Funding Number: TEA 1020-109.

**Institutional Review Board Statement:** The study was conducted in accordance with the Declaration of Helsinki and approved by the Institutional Review Board (Ethics Committee) of the University of Strathclyde (Details of protocol available at: https://www.strath.ac.uk/ethics/, accessed on 18 April 2022). Date of approval available on request.

**Informed Consent Statement:** Informed consent was obtained from all subjects involved in the study.

**Data Availability Statement:** The data presented in this study are not available as the permissions for public availability of interview transcripts were not given by the interviewees.

**Acknowledgments:** The study is part of an ongoing PhD research on TDC for the NBD of the SBE. The author would like to thank all interviewees and informants who participated in the research in the first year of doctoral studies.

**Conflicts of Interest:** The authors declare no conflict of interest.

## Appendix A. Summary of Responses by Academics from FoE and FoS, University of Strathclyde

*Appendix A.1. Inspiration*

Two responses (A6, A72) were received regarding inspiration for NBD applied in SBD. Bio-inspired elements in a design play a role in structural integrity and gave several examples that are translatable into buildings, such as honeycomb structures, the birds' wings, and the bone's structure (A6). The same participant referred to Barcelona, Spain, as a city with several bio-inspired buildings designed by Antoni Gaudí. Further discussion provided the input that NBD may help explore new materials related to structural bracing, building insulation, etc. If a designer needs inspiration from an animal or plant to implement a biologically inspired process that could inform an engineering solution, then they may need formal education in biology and zoology (A6).

The second academic (A72) suggested that biological interactions contribute to the SBE if we can control molecules at the nanoscale and extrapolate it to an architectural level. Optically responsive nanomaterials are an inspiration for architecture for the design of stained-glass windows. These materials allow changing the appearance of a building by covering its facades to make it visually attractive or by harvesting energy from sunlight to instigate a color change. Studying biological interactions allows creating novel nanostructures; however, synthetic materials are vastly simplified and simpler than what exists in nature. The availability of materials used to be a major barrier because biological materials were costly and seldom available; however, today complexity is the main problem (A72). The structure of seashells was presented as an example of natural, well-defined crystalline material with extensive molecular detail. These structures may draw some analogy for the built environment due to their sheer strength applicable to buildings.

*Appendix A.2. Design*

Commenting on design, the respondent (A10) suggested the use of biomimetic surface coatings to improve building performance. Biomimetic materials for buildings could mimic insect skins and control aspects such as transparency and texture for energy management. Thin surface coatings could make building materials and buildings more sustainable (A10). Biomimicry specialists do not comprehensively examine building materials to achieve their whole lifecycle sustainability. Nature and plant-based materials for building design have the potential to be sustainable; however, this aspiration has been limited due to reduced availability, focus on functionality, and financial costs. These limitations have created tensions between function, sourcing nature-based or plant-based material, and total cost competition (A10). Interestingly, there was a comment that it is uncertain whether a sustainable material will have enhanced functionality.

*Appendix A.3. Materials*

The highest number of responses (A11, A54, A70) were related to the theme of materials. The first respondent (A70) explained that looking into nature for potential applications in architecture or science allows the discovery of intriguing effects, working principles, and efficient solutions that have evolved over millions of years to solve complicated problems. These effects, principles, and solutions could be abstracted and applied to technical fields to solve complex problems and drive innovation forward. The interviewee highlighted the "lotus effect" and its significance for innovating self-cleaning materials for sustainable buildings. The challenge for developing these materials is to seek knowledge of working mechanisms in the biological world. It was suggested that biologists need to comprehensively enhance the existing biological knowledge so architects, chemists, physicists, and engineers can use it. Other identified challenges include the lack of understanding of nature's multiparameter space, its mechanisms, and the interrelation of its counterparts.

The second researcher (A11) focused on the development of sustainable materials to move away from fossil fuels. The participant pointed out that energy demands of manufacturing natural polymers should be lower than fossil-fuel-based plastic production, using less energy and contributing any waste to a circular economy. It was advocated that "essential plastics" in the construction industry need to be biodegradable or compostable to ensure whole lifecycle sustainability. The identified challenges related to materials included high-energy lab and material costs, and wasteful solvent disposal. The interviewee defined the concept of "true circular economy" as a process of converting materials back to their raw materials instead of merely upscaling or "greening" a part of the material manufacturing process. It was deduced that upscaling does not resolve issues related to end-of-life product sustainability.

The third participant (A54) highlighted the contribution of sustainably sourced nanocomposite materials to the SBE. There is a consensus among participants A54 and A10 as they suggested nanocomposite materials could be environmentally friendly but might not be completely sustainable as scientists aim for functionality that could conflict with sustainability. The researcher suggested that nature should be mimicked to reinforce materials, build complex structures, and construct nanostructures. It was proposed that NBD could develop sustainable nanocomposite materials if biological structural qualities are replicated. Challenges identified by the respondent include an extensive and complicated range of scales among the disciplines of chemistry, biology, and engineering for MDR as chemists use nanometers, biologists use micrometers, and engineers use millimeters and centimeters. The participant expressed their concern that these professionals never fully understand each other and stated that a chemist would understand finite element analysis but not its engineering aspects.

*Appendix A.4. Reuse*

Regarding the theme of reuse, the respondent (A2) agreed with A11 and suggested that a circular economy encourages reusing natural resources as it mimics natural processes that do not produce waste. NBD could generate a circular economy in the building industry and presented a few examples, such as reusing existing buildings in planning building developments, reusing building elements, and recycling building materials (A2). An identified barrier in applying a circular economy in the construction sector is the need for and cost of testing the performance of reused building elements and materials.

*Appendix A.5. Carbon Dioxide Reduction*

Two responses (A42, A45) were received concerning the theme of carbon dioxide reduction. The first researcher (A42) clarified that low-carbon technologies contribute to the sustainability of the built environment and gave an example from their experience in Brazil. It is one of the leading countries having an integrated energy matrix without hydro-energy, not so much solar energy but bioethanol coming from sugarcane (A42). After the Kyoto Protocol became effective in 2005, Brazil shifted from its dependence on oil and petroleum to renewable energy. The respondent stated that due to the planning of several hydro dams in Brazil, thousands of poor communities were displaced and were not provided with electricity or paid proportionally high energy costs. The accessibility and affordability of energy to local communities in planning renewable energy projects that use low-carbon technologies should be one of the aims to achieve sustainability. These concepts are interlinked, as the construction of low-carbon projects needs to consider local communities when identifying the available natural resources to achieve an optimum solution—solar, hydro, etc. The main barrier to successfully implement a low-carbon technology includes the lack of integration of social elements and limited engagement with the people they are designed to benefit. Ignoring social elements will remain a barrier until technological innovation engages with local societal needs, encourages people to get involved, and finds ways for people to help run these new technologies (A42). The participant stressed that if a new low-carbon technology is "parachuted on top of the communities", it has a risk

of failure as it would lack social engagement, and people would not invest in using them. It was deduced that for implementing NBD and low-carbon technologies for an energy transition promising a more sustainable future, there needs to be both ecological and social commitment to reduce carbon emissions and reduce inequality.

The second researcher (A45) agreed that effective design decisions need timely information to understand the built environment. There was a suggestion that pervasive monitoring allows collecting, collating, and delivering performance information (energy use, indoor conditions, and environmental impacts), and energy system simulation can provide realistic predictions of future performance. The academic confirmed that a combination of pervasive monitoring and energy-system simulation allows for the resilience testing of proposed designs and reinforces the design intent realized in practice. Modeling/monitoring tools need to be integrous to exhibit reality developed using theories and methods that are good representations of the thermo-fluid mechanisms governing all natural systems (A45). The significant barrier in such an integration is creating high-integrity modeling and monitoring tools requiring the development and validation of detailed algorithms for different aspects such as heat transfer, fluid flow, weather, and human behavior.

### *Appendix A.6. Research Methods*

Two responses (A12, A13) were received regarding research and education methods in the application of NBD for SBE. The first participant (A13) pointed out that multidisciplinary education of architects can contribute to the design of SBE. Several relevant topics were mentioned: green roofs, green infrastructure, environmental enhancement, the link to sustainable development goals, ecosystem services, circular economy in construction, new bioproducts replacing fossil-fuel-based products, and the use of BIM to help reuse materials available due to deconstruction. It was cautioned that just because some approaches are named as NBD, it does not mean that they would ensure reduced carbon emissions (A13).

The second researcher (A12) highlighted the need for multidisciplinary education of architects and that it is essential that students work in multidisciplinary teams to pick up ideas and develop diverse working strategies. By learning from nature and working together, architecture students can understand the influence of humans on the environment and how nature deals with complex problems (A12). Working in a multidisciplinary team allows the development of a broader perspective. The education system for a discipline generally operates in isolation, which is not reflective of what happens later in a workplace when it is necessary to collaborate with other disciplines and broker relationships to access information from their fields. The academic (A12) reinforced the insight of A13, suggesting that using a natural or sustainable material does not ensure carbon emission reduction and stressed that sometimes processing these materials is more energy-intensive and emits more carbon unless there is a sustainable energy source. Recommendations included that the correlation between NBD and carbon emission reduction needs to be taught, using multidisciplinary education. Achieving multidisciplinary education is challenging as departments are reluctant to implement it due to issues such as credit weighting, staff workload allocation, stakeholder management, difficult coordination, disagreement on ownership, and unbalanced sharing of responsibility (A12). Negotiations for this type of scheme tend to be very difficult. At the Faculty of Engineering at the University of Strathclyde, there is an emphasis on multidisciplinarity in engineering and physical sciences, but not so much in biological sciences. An example is the postgraduate program on Sustainable Engineering, common to all departments within the Faculty of Engineering, that has core elements that support multidisciplinary education.

### *Appendix A.7. Ecological Living*

Two responses (A27, A35) were received regarding ecological living and its relationship to NBD for the SBE. The first researcher (A27) confirmed that bio-based strategies improve rural lifestyle as they encourage using local resources such as agricultural products, forestry,

fishery, etc. Utilizing rural area potentials include planning facilities for IT servers at colder locations to reduce cooling costs, generating employment, improving rural lifestyle, and stopping demographic change (A27). The academic shared their experience from Switzerland and compared it to Scotland, and mentioned self-building ideas being more advanced and prevalent in the former. The respondent believes that self-building invites an NBD approach in residential areas of settlements, but is challenged by low awareness, planning restrictions, and legal and financial barriers. Finding professionals who are specialists in NBD is quite difficult; for example, an architect who specializes in zero-carbon housing or a biomimicry expert. It was mentioned that policies are important to support the design of sustainable architecture, e.g., subsidies to young families living in rural areas in exchange for complying with criteria for SBD.

The second researcher (A35) pointed out that smart systems in public transport, waste recycling, and vertical farming contribute to a sustainable urban environment, minimize environmental impact, and increase the quality of life of citizens. For achieving optimum design solutions that meet societal needs, new methodologies such as NBD are necessary to respond to complex highly interdependent urban systems. Major challenges include establishing a co-development environment comprising relevant experts, existing service providers, and citizens who would all be active participants and stakeholders in developing a solution and developing a "sense of ownership."

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
