# Peer review of "Multidisciplinary and Transdisciplinary Collaboration in Nature-Based Design of Sustainable Architecture and Urbanism"

_sustainability, doi:10.3390/su141610339_

Round 1

Reviewer 1 Report

The manuscript is well written and organized. The argument is original and aligned with the scope of the journal. According to my opinion the manuscript could be accepted for publication after minor improvements.

Section 3.3 – “Multidisciplinarity in education and research in building design”

In Figure 2 there is a graphic error in the upper right corner.

Section 4.1 - Survey findings

An introductory scheme for organizing the (summarized) contents presented (in full) in Table 3 could be useful for understanding all the areas in which this investigation has explored.

Section 5 – “Discussion”

What is proposed favours collaboration between academia and industry in eco-design. How, this contamination could also affect the teaching that the academy provides, as well as pure research. Consider for example: “Spreafico, C., & Landi, D. (2022). Using Product Design Strategies to Implement Circular Economy: Differences between Students and Professional Designers. Sustainability, 14(3), 1122.”  and “Spreafico, C., & Landi, D. (2022). Investigating students’ eco-misperceptions in applying eco-design methods. Journal of Cleaner Production, 342, 130866.”.

Author Response

Dear Reviewer,

Thank you very much for your useful comments, which facilitated the improvement of the paper. Please see the attachment on how each comment has been addressed (page numbers and line numbers are as per the print view).

Reviewer 2 Report

This manuscript contributes to current research through exploring multidisciplinary and transdisciplinary collaboration in nature-based design of sustainable built environment. However, this manuscript requires further improvement.

(1)  The Introduction should be reorganized. It is inappropriate to introduce nature-based design methods in a large part of the introduction. This part should focus on highlighting the introduction of scientific issues to prove the significanceof the research, such as the need for multidisciplinary and transdisciplinary collaboration in nature-based design of sustainable built environment, followed by a systematic review of the content, methods and scale of research about sustainable built environment and nature-based design in highly condensed language to prove the innovation of this research. Therefore, the introduction is untenable.

(2)  Lack of discussion on interdisciplinary research in Section 3.1. In this part, a lot of content about multidisciplinary research and transdisciplinary research has been mentioned. However, the discussion on interdisciplinary research is not sufficient.

(3)  The Section 3.3 can be integrated. Section 3.3 reviewed the relative studies about multidisciplinarity in education and research in building design, but we prefer to conduct a summary and systematic review of research rather than elaborate on existing findings.

(4)  The Section 3.4 needs improvement. Section 3.4 has discussed on multidisciplinary approaches and collaborative practices for sustainable architecture and urbanism from 7 perspectives through a large number of cases, but the length of individual aspects is not enough, such as buildings materials, vegetation on and in buildings, and infrastructures. In addition, a summary overview of the seven perspectives is lacking.

(5)  Section 4 has explored multidisciplinary approaches for NBD in engineering and science fields through a case study. The most significant survey findings have been introduced in Section 4.1, but there is no specific analysis of this part, which lacks the presentation of research conclusions. It also does not explain what significance these findings have for the research of this paper.

(6)  The abstract and research results of this paper both have mentioned the need to establish a transdisciplinary collaboration framework for research on NBD to support knowledge exchange within academia and with industry, but it does not mention what aspects should be considered in this framework, nor does it explain how SBE design and NBD should be reflected in this framework.

(7)  Supplement more literature of authoritative journals in this field.

Author Response

(The authors gave the same response as above.)

Reviewer 3 Report

The manuscript subject “Multidisciplinary and transdisciplinary collaboration in the nature-based design of sustainable architecture and urbanism” is interesting. It seems like a review paper, and the novelty and presented method are not enough for an original article.

1.      Figures 1 and 2 do not show any specific information and can be deleted.

2.      The challenges of multidisciplinary design can be written in one sentence instead.;

3.      In lines 40-41: The references number from [6] jumped to [50];

4.      Adding more new references especially those published in 2021, seems necessary in this paper.

5.      The case study analysis is about NBD related research by the people/academic members of the engineering and Science departments. However, in the abstract, a misunderstanding could happen that the case study analysis is about the previous NBD interventions in two departments not the published papers. The abstract needs to be re-written better.

6.      In line 93 is mentioned, “identified knowledge gap are presented in Section 3.” Adding identified knowledge gaps in one table at the end of this section seems necessary.

7.      Why are the authors limited the analysis of the published scientific papers about NBD to review the published papers in two departments of Strathclyde university?  How can this case study improve the analysis and the “knowledge gap” as mentioned in line 497?

8.      Section 4 is given up without any description of the results. Even in “4.1. Survey findings” there is no analysis of the presented table.

Author Response

(The authors gave the same response as above.)

Reviewer 4 Report

The paper "The impact of greenery and surface reflectance’s on solar radiation in perimeter blocks " is an original paper investigating multidisciplinary approaches to the design of sustainable built environment. The topic and research area of the study are interesting.

However, my general impression is that the case study in section 4 is detached from the literature review in section 3. The format of the whole paper looks strange and it is neither a literature review paper nor a case study paper. The literature review part looks too broad and general, and includes many aspects, such as the multidisciplinarity in education, infrastructures, smart cities, vegetation on and in buildings, Building materials , Building design and improvement. However, the case study just focus on the nature based design. The introduction section lacks the research gaps and contributions of this study. The contributions to the state-of-art based on the findings of the paper are not compelling from my eyes.

Author Response

(The authors gave the same response as above.)

Round 2

Reviewer 2 Report

This manuscript presented a detailed literature review to explore the applications of multidisciplinary research in building design and provided a particular case study to explain the challenges of multidisciplinary research in the application of nature-based design methods. This manuscript contributes to the development of a general framework for conducting transdisciplinary research in the future. This study has certain strengths and limitations. Therefore, some revisions are still needed. I have some specific suggestions and comments as follows:

1. The current literature review showed many case studies and the views of different scholars, but there are no generalized views and concluding remarks. The current literature review is a bit casual and loose. So, the literature review could be improved.

2. For the 4.1 Section, this analysis of Survey findings remains too superficial. L584-588 and Table 4 essentially show duplicated information. Greater concision would help the main messages stand out by avoiding repetition.

3. The structure of this manuscript is unclear, for example, the 4.2 Section is disappeared. And I think this manuscript can add a Conclusion section to separate the Discussion section and Conclusion section.

4. The manuscript contains some spelling errors, for example, ‘cahallenges’ in L14. The full text needs to be checked more carefully to avoid such errors.

Author Response

Dear Reviewer,

Thank you very much for your useful comments, which facilitated further improvement of the paper.

Please see the attachment on how each comment has been addressed (page numbers and line numbers are as per the print view).

Reviewer 3 Report

I have no further comments.

Author Response

Dear Reviewer,

Thank you very much for your useful comments, which facilitated further improvement of the paper.

Please see the attachment on how each comment has been addressed.

Reviewer 4 Report

Authors tried to make the structure and logical coherence of the study more clear. The quality of paper has been improved a lot.

Author Response

(The authors gave the same response as above.)
